# REFACTOR: Learning to Extract Theorems from Proofs

**Jin Peng Zhou** [*†]
Cornell University

**Yuhuai Wu** [*]
xAI

**Qiyang Li**
University of California, Berkeley

**Roger Grosse**
University of Toronto
Vector Institute

## Abstract

Human mathematicians are often good at recognizing modular and reusable theorems that make complex mathematical results within reach. In this paper, we propose a novel method called theoREm-from-prooF extrACTOR (REFACTOR) for training neural networks to mimic this ability in formal mathematical theorem proving. We show on a set of unseen proofs, REFACTOR is able to extract $19.6\%$ of the theorems that humans would use to write the proofs. When applying the model to the existing Metamath library, REFACTOR extracted 16 new theorems. With newly extracted theorems, we show that the existing proofs in the MetaMath database can be refactored. The new theorems are used very frequently after refactoring, with an average usage of 733.5 times, and help shorten the proof lengths. Lastly, we demonstrate that the prover trained on the new-theorem refactored dataset proves more test theorems and outperforms state-of-the-art baselines by frequently leveraging a diverse set of newly extracted theorems. Code can be found at https://github.com/jinpz/refactor.

## 1 Introduction

In the history of calculus, one remarkable early achievement was made by Archimedes in the 3rd century BC, who established a proof for the area of a parabolic segment to be $4/3$ that of a certain inscribed triangle. In the proof he gave, he made use of a technique called the *method of exhaustion*, a precursor to modern calculus. However, as this was a strategy rather than a theorem, applying it to new problems required one to grasp and generalize the pattern, as only a handful of brilliant mathematicians were able to do. It wasn't until millennia later that calculus finally became a powerful and broadly applicable tool, once these reasoning patterns were crystallized into modular concepts such as limits and integrals.

A question arises – can we train a neural network to mimic humans' ability to extract modular components that are useful? In this paper, we focus on a specific instance of the problem in the context of theorem proving, where the goal is to train a neural network model that can discover reusable theorems from a set of mathematical proofs. Specifically, we work under formal systems where each mathematical proof is represented by a tree called *proof tree*. Moreover, one can extract some connected component of the proof tree that constitutes a proof of a standalone theorem. Under this framework, we can reduce the problem to training a model that solves a binary classification problem where it determines whether each node in the proof tree belongs to the connected component that the model tries to predict.

To this end, we propose a method called theoREm-from-prooF extrACTOR (REFACTOR) for mimicking humans' ability to extract theorems from proofs. Specifically, we propose to reverse the process of human theorem extraction to create machine learning datasets. Given a human proof $P$, we take a theorem $t$ that is used by the proof. We then use the proof of theorem $t$, $P_t$, to re-write $P$ as $P'$ such that $P'$ no longer contains the application of theorem $t$, and replace it by using the proof $P_t$. We call this re-writing process the *expansion* of proof $P$ using $t$. The expanded proof $P'$ becomes the input to our model, and the model's task is to identify a connected component within $P'$, $P_t$, which corresponds to the theorem $t$ that humans would use in $P$. Please see Figure 1 for visualization.

---

[*]Equal contribution. Work done while at University of Toronto and Vector Institute.
[†]Correspondence to: Jin Peng Zhou <jpzhou@cs.cornell.edu>

We implement this idea within the Metamath theorem proving framework – an interactive theorem proving assistant that allows humans to write proofs of mathematical theorems and verify the correctness of these proofs. Metamath is known as a lightweight theorem proving assistant, and hence can be easily integrated with machine learning models (Whalen, 2016; Polu & Sutskever, 2020). It also contains one of the largest formal mathematics libraries, providing sufficient background for proving university-level or Olympiad mathematics. Our approach is also generally applicable to other formal systems such as Lean (de Moura et al., 2015), Coq (Barras et al., 1999) and HOL Light (Harrison, 1996) since proofs in these environments can also be represented as trees and mathematically support the substitution of lemmas. Moreover, our approach could go beyond theorem proving and be implemented in program synthesis by inlining expansion. In this paper, we instead focus on theorem proving to mechanize mathematical proofs. we chose Metamath for this project because it is simplest to work with as it only has one tactic (inference rule) – "substitution".

Unlike previous methods that are mostly symbolic Vyskočil et al. (2010) or mining-based Kaliszyk et al. (2015), we propose a more generic approach, that is the first training a neural network to extract useful lemmas from proofs. Our best REFACTOR model is able to extract exactly the same theorem as humans' ground truth (without having seeing instances of it in the training set) about $19.6\%$ of time. We also observe that REFACTOR's performance improves when we increase the model size, suggesting significant room for improvement with more computational resources.

Ultimately, the goal is not to recover known theorems but to discover new ones. To analyze those cases where REFACTOR's predictions don't match the human ground truth, we developed an algorithm to verify whether the predicted component constituent a valid proof of a theorem, and we found REFACTOR extracted 1907 valid, new theorems. We also applied REFACTOR to proofs from the existing Metamath library, from which REFACTOR extracted another 16 novel theorems. Furthermore, with newly extracted theorems, we show that the existing theorem library can be refactored to be more concise: the extracted theorems reduce the total size by approximately 400k nodes. (This is striking since REFACTOR doesn't explicitly consider compression as an objective). Lastly, we show in Table 4 that training a prover on the refactored dataset leads to proving 75 more test theorems, outperforming a state-of-the-art baseline, MetaGen (Wang & Deng, 2020). Out of all proved test theorems, there are $31.0\%$ of them that use the newly extracted theorems at least once. The usages span across $141$ unique newly extracted theorems, further suggesting diverse utility in new theorems we extracted.

Our main contributions are as follows: 1. We propose a novel method called REFACTOR to train neural network models for the theorem extraction problem, 2. We demonstrate REFACTOR can extract unseen human theorems from proofs with a nontrivial accuracy of $19.6\%$, 3. We show REFACTOR is able to extract frequently used theorems from the existing human library, and as a result, shorten the proofs of the human library by a substantial amount. 4. We show new-theorem refactored dataset can improve baseline theorem prover performance significantly with newly extracted theorem being used frequently and diversely.

## 2 RELATED WORK

**Lemma Extraction**   Our work is generally related to lemma mining in Vyskočil et al. (2010); Hetzl et al. (2012); Gauthier & Kaliszyk (2015); Gauthier et al. (2016) Rawson et al. (2023) and mostly related to the work of Kaliszyk & Urban (2015); Kaliszyk et al. (2015). The authors propose to do lemma extraction on the synthetic proofs generated by Automated Theorem Provers (ATP) on the HOL Light and Flyspeck libraries. They showed the lemma extracted from the synthetic proofs further improves the ATP performances for premise selection. However, previous methods cannot be directly applied to our problem since they rely on feature engineering with large overhead. Algorithms such as PageRank that ranks existing theorems and lemmas are also not applicable since our goal is to discover and extract new theorems.

**Discovering Reusable Structures**   Our work also is related to a broad question of discovering reusable structures and sub-routine learning. One line of the work that is notable to mention is the Explore-Compile-style (EC, EC2) learning algorithms  (Dechter et al., 2013; Ellis et al., 2018; 2020; Bowers et al., 2023). These works focus on program synthesis while trying to discover a library of subroutines. As a subroutine in programming serves a very similar role as a theorem for theorem

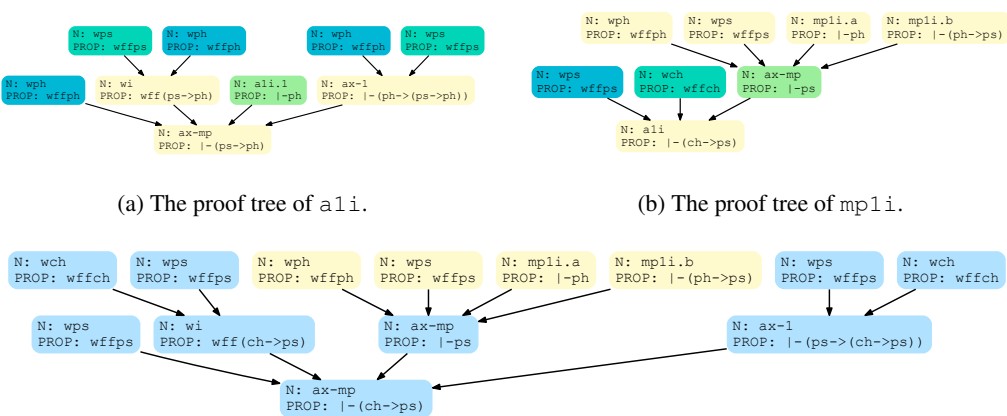

(a) The proof tree of `ali`.  (b) The proof tree of `mp1i`.

(c) The proof tree of `mp1i` with theorem `ali`'s proof substituted (colored in blue).

Figure 1: (a) and (b): proof tree visualization of theorems `ali` and `mp1i` respectively. Each node of the proof tree contains two pieces of information: N and PROP. N refers to the name of the premise, axiom or theorem applied at this node and PROP is the resultant expression after applying N. Note that in (a), `ali` has three hypotheses which are colored darkblue, darkgreen and lightgreen. In (b), proof of `mp1i` invokes theorem `ali` and the three corresponding hypotheses to theorem application are highlighted with the same color. (c): proof tree of `mp1i` in (b) is expanded by substituting the proof tree of `ali` in blue. These blue nodes, $\mathcal{V}_{target}$, are the targets for our proposed learning task.

proving, their work is of great relevance to us. However they approach the problem from a different angle: they formalize sub-routine learning as a compression problem, by finding the best subroutine that compresses the explored solution space. However, these works have not yet been shown to be scalable to realistic program synthesis tasks or theorem proving. We, on the other hand, make use of human data to create suitable targets for subroutine learning and demonstrate the results on realistic formal theorem proving. Another related line of work builds inductive biases to induce modular neural networks that can act as subroutines (Andreas et al., 2015; Gaunt et al., 2017; Hudson & Manning, 2018; Mao et al., 2019; Chang et al., 2019; Wu et al., 2020; Ito et al., 2022; Hersche et al., 2023). These works usually require domain knowledge of sub-routines for building neural architectures and hence not suitable for our application.

**Machine Learning for Theorem Proving**  Interactive theorem provers have recently received enormous attention from the machine learning community as a testbed for theorem proving using deep learning methods (Bansal et al., 2019a;b; Gauthier et al., 2018; Huang et al., 2019; Yang & Deng, 2019; Wu et al., 2021; Li et al., 2021; Polu & Sutskever, 2020; Aygün et al., 2022; Nawaz et al., 2021; Yang et al., 2023). Previous works demonstrated that transformers can be used to solve symbolic mathematics problems (Lample & Charton, 2020), capture the underlying semantics of logical problems relevant to verification (Hahn et al., 2020), and also generate mathematical conjectures (Urban & Jakubův, 2020). Rabe et al. (2020) showed that self-supervised training alone can give rise to mathematical reasoning. Li et al. (2021) used language models to synthesize high-level intermediate propositions from a local context. Piotrowski & Urban (2020) used RNNs to solve first-order logic in ATPs. Wang et al. (2020) used machine translation to convert synthetically generated natural language descriptions of proofs into formalized proofs. Yang & Deng (2019) augmented theorem prover with shorter synthetic theorems which consist of arbitrary steps from a longer proof with maximum length restriction. This is remotely related to our work where our extraction does not have such restrictions and instead attempts to learn from targets derived from human-written theorems.

## 3 METAMATH AND PROOF REPRESENTATION

In this section, we describe how one represents proofs in the Metamath theorem proving environment. We would like to first note that even though the discussion here specializes in the Metamath

environment, most of the other formal systems (Isabelle/HOL, HOL Light, Coq, Lean) have very similar representations. In the seminal work by Curry (1934); Howard (1980); Wadler (2015), the equivalence of proofs and computation trees are established. We refer readers to these work for a more formal treatment and we provide a high-level intuition specific to Metamath environment here. The fundamental idea is to think of a theorem as a function, and the proof tree essentially represents an abstract syntax tree of a series of function applications that lead to the intended conclusion.

Proof of a theorem in Metamath environment is represented as a tree. For example, the proof of the theorem `a1i` is shown in Figure 1 (a). Each node of the tree is associated with a *name* (labeled as `N`), which can refer to a premise of the theorem, an axiom, or a proved theorem from the existing theorem database. Given such tree, one can then traverse the tree from the top to bottom, and iteratively prove a true proposition (labeled as `PROP`) for each node by making a step of *theorem application*. The top-level nodes usually represent the premises of the theorem, and the resulting proposition at the bottom node matches the conclusion of the theorem. In such a way, the theorem is proved.

We now define one step of theorem application. When a node is connected by a set of parent nodes, it represents one step of theorem application. In particular, one can think of a theorem as a function that maps a set of hypothesis to a conclusion. Indeed, a node in the tree exactly represents such function mapping. That is, the node maps the set of propositions of its parent nodes, to a new conclusion specified by the theorem. Formally, given a node $c$ whose associated name refers to a theorem $T$, we denote its parent nodes as $\mathcal{P}_c$. We can then prove a new proposition by applying the theorem $T$, to all propositions proved by nodes in $\mathcal{P}_c$.

The proof of the theorem `a1i` in Figure 1 (a) consists of 3 theorem applications. In plain language, the theorem is a proof of the fact that if `ph` is true, then `(ps->ph)` is also true. The top-level nodes are the hypotheses of the theorem. Most of the hypotheses state that an expression is a well-formed formula so that the expression can be used to form a syntactically correct sentence. The more interesting hypothesis is `a1i.1`, which states `|-ph`, meaning `ph` is assumed to be true. In the bottom node, the theorem invokes the theorem `ax-mp`, which takes in four propositions as hypotheses, and returns the conclusion `|-(ps->ph)`. The entire proof can be thought as a function that takes in three arguments: `wffph`, `wffps` and `|-ph`, and outputs `|-(ps->ph)`.

## 4 METHOD

In this section, we describe our approach to training neural network models for extracting useful theorems from proofs. As one can represent mathematical proofs as trees, we first discuss how to identify a connected subtree of the proof tree with a valid proof of another theorem. We then formalize the problem of theorem extraction as a node-level binary classification problem on the proof tree. Next, we propose an algorithm that expands a theorem's proof inside of another proof, to create suitable targets for learning theorem extraction. Finally, we give an algorithm that verifies if the component predicted by the model constitutes a valid proof of a theorem, and if so, turns the component into a theorem.

### 4.1 SUB-COMPONENT OF A PROOF TREE AS A THEOREM

We have discussed how one can represent a mathematical proof as a proof tree in section 3. Interestingly, one can also identify some components of the proof tree with an embedded proof of another theorem. To start with, given a node in a proof tree, one can treat the entire subtree above that node as a proof of the node (more precisely, the proposition contained in the node, i.e., `PROP`). For example, in the proof of `a1i` in Figure 1 (a), the subtree above the node `ax-1` consists of two hypotheses `wffph` and `wffps`, and they constitute a proof of the proposition `|-(ph->(ps->ph))` contained in the node `ax-1`.

In addition to the entire subtree above a node, one may identify some connected subtree of the tree with a valid theorem. For example, in Figure 1 (c), we show that the proof of the theorem `mp1i` contains an embedded proof of the theorem `a1i`. The embedded proof is colored in blue, and there is a one-to-one correspondence between these blue nodes and the nodes in the proof of `a1i` shown in Figure 1 (a). One can hence refactor the proof with an invocation of the theorem `a1i`, resulting in a much smaller tree shown in Figure 1 (b).

In general, a connected subtree is only a necessary condition and there are more criteria a component needs to satisfy to be identified as a valid proof of a theorem. In Appendix A.1, we develop such an

algorithm in more detail that performs the verification for theorem extraction. We will use that to verify the prediction given by a neural network model.

To conclude, in this section, we establish the equivalence between theorem extraction from a proof as to the extraction of a sub-component from a proof tree. This allows us to formalize the problem as a node-level prediction problem on graphs as we introduce next.

## 4.2 SUPERVISED PREDICTION TASK

The model is given a proof tree $\mathcal{G}$ with a set of nodes $\mathcal{V}$, edges $\mathcal{E}$, and node features $x_v$ which correspond to the name N and the proposition PROP associated with each node. The task is to output a subset of nodes $\mathcal{V}_{\text{target}} \subset \mathcal{V}$ that correspond to an embedded proof of a useful theorem. We cast the problem as a node-level binary classification problem that predicts whether each node belongs to $\mathcal{V}_{\text{target}}$. Without loss of generality, we let all nodes in $\mathcal{V}_{\text{target}}$ to have labels of 1 and the rest 0.

We use a graph neural network parametrized by $\theta$ to take a single graph and its node features as input, and outputs a scalar $\hat{P}_v$ between 0 and 1 for each node $v \in \mathcal{V}$, representing the probability belonging to $\mathcal{V}_{\text{target}}$. Our objective is a binary cross entropy loss between the predicted node level probabilities and the ground truth targets for a graph. Because the number of nodes usually varies significantly across proofs, we normalize the loss by the number of nodes in the graph[1]:

$$\mathcal{L}(G, \theta) = -\frac{1}{|\mathcal{V}|} \sum_{v \in \mathcal{V}_{\text{target}}} \log P(\hat{P}_v = 1 | \mathcal{G}, \theta) \tag{1}$$

$$-\frac{1}{|\mathcal{V}|} \sum_{v \notin \mathcal{V}_{\text{target}}} \log P(\hat{P}_v = 0 | \mathcal{G}, \theta) \tag{2}$$

We then seek the best parameters by minimizing the loss over all proof trees:

$$\arg\min_\theta \sum_G \mathcal{L}(G, \theta). \tag{3}$$

## 4.3 REFACTOR: THEOREM-FROM-PROOF EXTRACTOR

With the prediction task formulated, we now describe how to generate training data points of proof trees $\mathcal{G}$ with suitable targets $\mathcal{V}_{\text{target}}$ defined. Even though we specialize our discussion in the context of Metamath, the same technique can be applied to other formal systems for creating datasets of theorem extraction, such as Lean (de Moura et al., 2015), Coq (Barras et al., 1999) and HOL Light (Harrison, 1996).

It is worth noting that even though the existing human proofs from the Metamath library cannot be used directly, they offer us hints as to how to construct training data points. To illustrate, in Figure 1 (b), the proof of mp1i invokes a theorem application a1i with three arguments (wffps, wffch, |-ps in darkblue, darkgreen and lightgreen respectively), which is a theorem that human considered useful and stored in the library. Our idea is to reverse the process of theorem extraction, by expanding the proof of a1i in the proof of mp1i to obtain a synthetic proof shown in 1 (c). In this expanded proof of mp1i, one can see the proof of a1i is embedded as a connected subtree colored in blue, hence creating a suitable target for theorem extraction.

We explain how we perform the proof expansion in detail. We think of the theorem as a function whose arguments are a set of hypotheses and the output is a conclusion, as mentioned in Section 3. Instead of calling the theorem by its name, we intentionally duplicate the body of its proof tree, and replace their canonical arguments with the arguments we wish to pass in context. There are three key steps: 1. identifying the proof tree associated with the theorem (e.g., a1i in Figure 1 (a)), substituting the original arguments with the actual ones in the proof context (e.g., substituting leaf nodes wffph in darkblue, wffps in darkgreen and |-ph in lightgreen in Figure 1 (a) with nodes wffps, wffch and |-ps in Figure 1 (b) of the same color respectively[2]), and finally copy and replace it to where the expanded node is located (e.g, replace a1i node in Figure 1 (b) with the

---

[1]In our preliminary experiments we found that the normalized loss gave better performance than weighting all nodes in the database equally.

[2]Note that these three nodes in Figure 1 (b) are parents, namely, arguments to a1i node in Figure 1 (b).

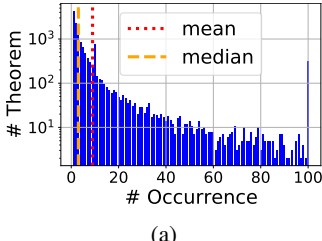 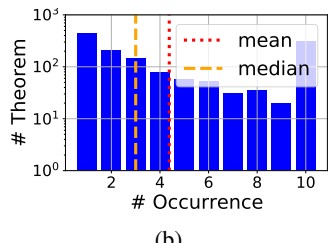 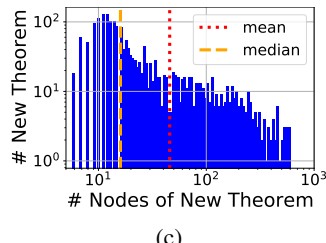

(a) (b) (c)

Figure 2: Number of theorems vs number of occurrences in entire dataset (a) and test set (b). Both (a) and (b) show noticeable occurrence unbalance with (b) being less due to our further subsampling of a maximum 10 occurrence for test set. (c) Distribution of number of nodes in newly extracted theorems. The model mostly extracts short theorems but is also capable of extracting theorems that have hundreds of nodes.

Table 1: Node level and proof level accuracy of REFACTOR with different input configurations. **No edge**: all the edges in the graph are removed; **Leaves→Root**: only keep the edges that are in the same direction of the paths that go from leaves to their parents; **Leaves←Root**: same as Leaves→Root except all the edges are reversed; **Leaves↔Root**: the original graph with bidirectional edges. **Node Features**: whether or not the node features are fed as input to the model. All the experiments are run with $K = 10$ and $d = 256$.

|  | Training Node Accuracy | Training Proof Accuracy | Test Node Accuracy | Test Proof Accuracy |
|---|---|---|---|---|
| No edge + Node Features | 86.8% | 0.1% | 74.9% | 0.1% |
| Leaves→Root + Node Features | 87.1% | 0.5% | 75.2% | 0.1% |
| Leaves←Root + Node Features | 96.6% | 6.0% | 88.1% | 3.5% |
| Leaves↔Root | 86.3% | 0% | 74.2% | 0% |
| Leaves↔Root + Node Features (**REFACTOR**) | 97.5% | 37.5% | 84.3% | 13.3% |

substituted `ali` to arrive at Figure 1 (c)). We present a more formal and detailed exposition of the algorithm in Appendix A.2.

Lastly, note that there are many options for theorem expansion. Firstly, one single proof can contain multiple theorems, and each theorem can be expanded either simultaneously or one by one. In addition, one can even recursively expand theorems by expanding the theorem inside of an expanded proof. For simplicity, in this work we only separately expand one theorem at a time. Hence, for a proof that contains $M$ total number of theorem applications, we create $M$ data points for learning theorem extraction. We leave investigations of more sophisticated expansion schemes to future work.

## 5 EXPERIMENTS

In this section, we evaluate the performance of REFACTOR via a variety of experiments. We begin by describing our dataset and experimental setup and then analyze the results to address the following research questions:

- **Q1**: How does REFACTOR perform when evaluating against ground truth theorem under a variety of ablations of data and model architectures?
- **Q2**: Are newly extracted theorems by REFACTOR used frequently?
- **Q3**: With newly extracted theorems, can we (a) compress the existing theorem library and (b) improve theorem proving?

### 5.1 DATASET AND PRE-PROCESSING

We applied REFACTOR to create datasets from the main and largest library of Metamath, `set.mm`. In order to fairly compare prover performance reported from Whalen (2016), we used their version of `set.mm`, which contains 27220 theorems. We also filtered out all expanded proofs with more than 1000 nodes or contain nodes features of character length longer than 512. This gave rise to 257264 data points for training theorem extraction before theorem maximum occurrence capping, which we describe next.

We noted that the distribution of theorem usage in `set.mm` is highly unbalanced. To prevent the model from learning to only extract a few common theorems due to their pervasiveness, we employed

a subsampling of the data with respect to theorem occurrence to balance the dataset. Specifically, in the training set, for those theorems that occur more than 100 times as extraction targets, we subsampled 100 data points per theorem. In Figure 2 (a), we plot a histogram of theorem occurrence versus the number of theorems. As seen in the figure, the distribution roughly follows a power-law distribution with 4000 theorems only used once in set.mm, and a substantial number of theorems that occur beyond 100 times. For the validation and test set, as we wanted to evaluate the model on a diverse set of extraction targets, we capped the maximum number of occurrences as 10 using subsampling. The occurrence histogram of the test dataset is shown in Figure 2 (b) and the total number of expanded proofs in our dataset after capping theorem maximum occurrence is 124294.

To evaluate the model's generalization ability, we performed a target-wise split on the dataset. That is, we split the dataset in a way that the prediction targets, namely, the theorems to be extracted, are non-overlapping for the train, valid and test set. By doing so, we discouraged extraction by simple memorization of common theorems.

## 5.2  MODEL ARCHITECTURE AND TRAINING PROTOCOL

In this section, we describe our neural network architecture parameters and other training details. We used a character-level tokenization for the node feature, which is a concatenation of texts in the fields N and PROP (see Figure 1). For each node, we first embedded all the characters with an embedding matrix, followed by two fully connected layers. We then averaged over all embeddings to obtain a vector representation of a node. We used these vector representations as the initial node embeddings to a graph neural network. We used $K$ GraphSage convolution (Hamilton et al., 2017) layers with size $d$ and two more fully connected layers with sigmoid activation at the end to output the scalar probability. The size of the character embedding was set to 128 and the number of hidden neurons in all the fully connected layers was set to 64. Both $K$ and $d$ are hyperparameters.

For model training, we used a learning rate of 1e-4 with Adam optimizer (Kingma & Ba, 2015). All methods were implemented in Pytorch Paszke et al. (2019) and Pytorch Geometric library Fey & Lenssen (2019). We ran all experiments on one NVIDIA Quadro RTX 6000, with 4-core CPUs.

## 5.3  Q1 - HOW MANY HUMAN-DEFINED THEOREMS DOES THE MODEL EXTRACT?

The results are summarized in Table 2. On the theorem extraction dataset obtained from Section 5.1, REFACTOR was able to correctly classify 85.6% of the nodes (Node Accuracy). For 19.6% (Proof Accuracy) of the proofs, REFACTOR was able to correctly classify all of the nodes and fully recover the theorem that the human use. We also show our approach scales well with the model size. As we increase the number of parameters by around 50x from 80k to 4M, both node and proof accuracy improve. In particular, the proof accuracy goes up significantly from 2.3% to 19.6%. This shows promise that the accuracy can be further improved with a larger model and larger dataset.

To understand what mechanism in the GNN made the theorem extraction possible, we re-trained the model, but with different configurations compared to the original training procedure. In particular, we examined the case where all the edges are removed (No edge) as well as two types of uni-directional connections: 1) only edges that go from leaves to root are included (Leaves→Root) and 2) only edges that go from root to leaves are included (Leaves←Root). In addition, we were curious to see whether the graph structure alone is sufficient for theorem prediction when no node features are provided.

We summarize the results of these configurations in Table 1 and report node level and proof level accuracy on training and test set. It can be seen that both edge connection and input node feature information are crucial in this task as both (No edge + Node Features) and (Leaves↔Root) achieved minimum proof level accuracy. Interestingly, the direction of edge led to a drastically different performance. Leaves→Root + Node Features performs poorly in proof level accuracy whereas Leaves←Root + Node Features achieved comparable performance with bidirectional edges (Leaves↔Root + Node Features).

The results can be explained by recognizing the fact that there are many identical hypothesis nodes in a proof due to MetaMath's low-level nature. For example, there are three identical leaf nodes wps in Figure 1 (c). If the edges only point from hypothesis to conclusion, the message for two identical hypothesis leaves will always be the same due to no incoming messages. Hence, it is theoretically impossible to make correct predictions on the proof level. On the other hand, the opposite direction of edges does not suffer from this limitation as there is only one root in the proof tree.

Table 2: Node level and proof level accuracy of REFACTOR with various model sizes.

| $K, d$, Number of Trainable Parameters | Training Node Accuracy | Training Proof Accuracy | Test Node Accuracy | Test Proof Accuracy |
|---|---|---|---|---|
| 5, 64, 80k | 89.4% | 5.1% | 77.4% | 2.3% |
| 5, 128, 222k | 91.3% | 9.9% | 78.6% | 3.0% |
| 5, 256, 731k | 93.7% | 17.3% | 80.1% | 4.4% |
| 10, 256, 1206k | 97.5% | 37.5% | 84.3% | 13.3% |
| 10, 512, 4535k | 97.9% | 42.7% | 85.6% | 19.6% |

Table 3: An analysis of incorrect predictions on the theorem extraction dataset.

| Dataset | Total | Not Tree & Invalid | Tree & Invalid | Tree & Valid |
|---|---|---|---|---|
| Training | 64349 | 13368 | 47521 | 3460 |
| Validation | 4766 | 1175 | 3238 | 353 |
| Test | 4822 | 1206 | 3348 | 328 |
| set.mm | 22017 | 8182 | 13470 | 365 |

Table 4: Number of test theorems proved comparison out of total 2720 test theorems.

| Setting | Test Proof Found | New Theorem Usage |
|---|---|---|
| Holophrasm'20 (Wang & Deng, 2020) | 557 | - |
| Symbolic Baseline (Vyskočil et al., 2010) | 566 | - |
| MetaGen (Wang & Deng, 2020) | 600 | - |
| REFACTOR (ours) | **632** | 31.0% |

As mentioned in Section 2, previous symbolic baselines are not directly applicable to our setting. Instead, we adapted and compared REFACTOR with a symbolic compression baseline that is similar to Vyskočil et al. (2010). The most frequently extracted theorems for the baseline achieves a 1.7% accuracy compared to 19.6% from REFACTOR. For implementation details, please see Appendix B.

## 5.4 Q2 - ARE NEWLY EXTRACTED THEOREMS BY REFACTOR USED FREQUENTLY?

In this section, we investigate whether theorems extracted by REFACTOR are used frequently. We used the best model (i.e., the largest model) in Table 2 for the results analyzed in this section. We explored two ways of extracting new theorems. We first investigated the incorrect predictions of REFACTOR on the theorem extraction dataset. When the prediction differs from the ground truth, it can correspond to a valid proof. We also applied REFACTOR on the human proofs of nodes less than 5000 from the library set.mm.

The number of valid theorems extracted from the theorem extraction dataset and set.mm are listed under *Tree & Valid* in Table 3. We observe that there were a non-trivial amount of predictions that led to valid theorems. Remarkably, we see REFACTOR was able to extract valid theorems in the real human proofs (set.mm), despite the fact that human proof distribution may be very different from the training distribution. Adding up all extracted theorems from both approaches, we arrived at 4204 new theorems. We notice that among them, some new theorems were duplicates of each other due to standardization and we kept one copy of each by removing all other duplicates. We also removed 302 theorems extracted on set.mm that corresponded to the entire proof tree. In the end, we were left with 1923 unique new theorems with 1907 and 16 from the expanded and original dataset respectively. We showed examples of extracted new theorems in the Appendix D.1. We also plot the distribution of number of proof nodes of the extracted theorems in Figure 2 (c). We can see the newly extracted theorems are of various sizes, spanning almost two orders of magnitudes, with some very sophisticated theorems that consist of hundreds of nodes.

We then computed the number of usages in set.mm for each newly extracted theorem, reported in Table 5. The average number of uses is 83 times, showing nontrivial utility of these theorems. Notably, the theorems extracted on set.mm are even more frequently used – 733.5 times on average. We think that because the human library is already quite optimized, it is harder to extract new theorems from existing proofs. But a successful extraction is likely to be of higher quality as the proof tree input represents a true human proof rather than a synthetically expanded proof.

We additionally performed a more detailed analysis on the predictions, by classifying them into three categories. The first category is denoted by *Non-Tree & Invalid* where the prediction is a disconnected set of nodes and hence it is impossible to form a new theorem. In the second category *Tree & Invalid*, the prediction is a connected subtree and hence forming a sub-tree, but it still does not satisfy other requirements outlined in our algorithm description to be a valid proof of a theorem (see Section 4.1 and Appendix A.1). The last category *Tree & Valid* corresponds to a prediction that leads to an extraction of new theorem previously not defined by humans. We present the number of predictions for each category in Table 3. We noticed the model predicted a substantial amount of disconnected components. We hypothesize this may be because our current model makes independent node-level predictions. We believe an autoregressive model has a great potential to mitigate this problem by encouraging contiguity, a direction which we leave for future work.

Table 5: Theorem usage and their contribution to refactoring.

| | # Theorems Used | Total Usage | Average Usage | Max Usage | Average Number of Nodes Saved | Total Number of Nodes Saved |
|---|---|---|---|---|---|---|
| Expanded | 670 | 147640 | 77.4 | 60705 | 196.7 | 375126 |
| Original | 14 | 11736 | 733.5 | 8594 | 2025.8 | 32413 |
| Total | 684 | 159376 | 82.9 | 60705 | 211.9 | 407539 |

### 5.5 Q3A - HOW MUCH CAN WE COMPRESS EXISTING LIBRARY WITH EXTRACTED THEOREMS?

When the newly extracted theorems are broadly reusable, we would expect the proofs in the library could be shortened by using the new theorems as part of the proofs. In this paper, we consider a specific re-writing procedure, which alternates between 1) matching the extracted theorems against the proofs in the library and 2) replacing the matched proportion of the proofs with the application of the new theorems (See more details in Appendix A.3). We call this procedure the *refactoring* procedure and the resulting shortened proof the *refactored* proof.

With the 16 new extracted theorems from the original dataset, the new library obtained from refactoring was indeed smaller (See Table 5). These new theorems on average saved 2025.8 nodes which is an order of magnitude more than those from the expanded dataset (196.7 nodes). Nevertheless, this shows that extracted theorems from both expanded and human datasets are frequently used in refactoring the theorem library. In total, we were able to refactor 14092 out of 27220 theorems in the MetaMath database. This improvement in compression is striking, as REFACTOR didn't explicitly consider compression as an objective.

### 5.6 Q3B - ARE NEWLY EXTRACTED THEOREMS USEFUL FOR THEOREM PROVING?

We further demonstrated the usefulness of our new theorems with an off-the-shelf neural network theorem prover, Holophrasm (Whalen, 2016). We trained two Holophrasm provers, one with the original dataset and the other with the dataset augmented with the newly extracted and refactored proofs. We compared number of proofs found on a hold-out suite of test theorems. All hyperparameters of the prover were set to default values with the time limit of each proof search set to 5 minutes.

We used the implementation from Wang & Deng (2020) as the baseline, which proved more test theorems than the original implementation. Additionally, we compare with the symbolic baseline similar to Vyskočil et al. (2010). More details of baseline implementation can be found in Appendix B and C. We summarized results in Table 4. It can be seen that by training on the refactored dataset, the prover was able to prove 75 more test theorems, a more than 13% improvement from Holophrasm. Additionally, we compare REFACTOR with a state-of-the-art baseline on Metamath, MetaGen (Wang & Deng, 2020), that trains a generator to generate synthetic theorems and trains Holophrasm with both original and synthethic theorems. REFACTOR in total found 632 test proofs, outperforming MetaGen and improving significantly from Holophrasm. These results suggest that REFACTOR is useful in theorem proving. We choose not to compare with GPT-f (Polu & Sutskever, 2020) since it uses very large transformers and significantly more compute. We leave potential combination of GPT-f and MetaGen with REFACTOR as our future work.

To investigate how newly extracted theorems contributed to the improvement, we calculated the percentage of proved theorems that used new theorem at least once in its proof (new theorem usage as shown in Table 4.) The usage is 31.0%, indicating newly extracted theorems were used very frequently by the prover. More remarkably, the newly extracted theorems used in proving test theorems did not concentrate on few theorems. Instead, there was a diverse set of newly extracted theorems that were useful in theorem proving: there were in total 141 unique new theorems used for proving test theorems, and the most frequently used one was used 17 times (see more details in Appendix D.2).

## 6 CONCLUSION

In this paper, we study extracting useful theorems from mathematical proofs in the Metamath framework. We formalize theorem extraction as a node-level binary classification problem on proof trees. We propose one way to create datasets and additionally develop an algorithm to verify the validity of the prediction. Our work represents the first proof-of-concept of theorem extraction using neural networks. We see there are various future work directions to improve the existing model such as using more powerful architectures like transformers to autoregressively predict the target. Lastly, we would like to note that our methodology is not only generic for formal mathematical theorem extraction, but also has the potential to be applied to other applications, such as code refactoring.

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

# A    FURTHER EXPLANATIONS OF THE ALGORITHMS

## A.1    THEOREM VERIFICATION

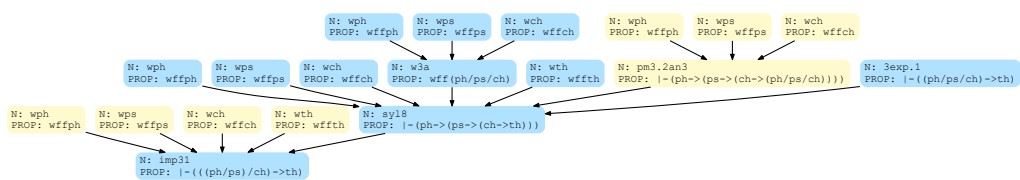

Figure 3: A proof tree prediction where nodes with output probability greater than 0.5 have been colored blue. This proof tree does not satisfy the constraint to be a valid theorem because only one of the parent nodes of the root (imp31) are predicted to be in $\mathcal{V}_{target}$.

In this section, we present our algorithm to determine whether a predicted component made by REFACTOR constitutes a valid theorem. On a high level, our algorithm checks for two necessary conditions and performs standardization before feeding the node names of extracted component to a verifier which we describe next.

We describe how we can verify Metamath proofs represented by a conclusion and a list of node names. We traverse the nodes following Reverse Polish Notation (RPN), and trace the theorem application results. When it finishes, we then compare the actual traced proposition given in the bottom node to the expected conclusion specified by the theorem. The proof is verified if and only if the two match with each other. We refer to this simple procedure as Metamath verifier.

For the theorem verification algorithm, We first extract all nodes whose prediction by our model is greater than 0.5, which we represent as $\hat{\mathcal{V}}_{target}$ (see Figure 4 (a) and (b)). We first check if $\hat{\mathcal{V}}_{target}$ forms a connected component i.e. a tree structure, as disjoint set of nodes cannot be a valid new theorem. Secondly, one necessary constraint for a valid extracted theorem is that for each node in $\hat{\mathcal{V}}_{target}$, either none or all of its parent nodes need to be present in $\hat{\mathcal{V}}_{target}$. If only some but not all parents are present, this corresponds to a step of theorem application with an incorrect number of arguments. We illustrate one example that violates this constraint in Figure 3. As seen in this example, only one parent of the root node imp31 is in $\hat{\mathcal{V}}_{target}$ and similarly one parent node of syl8 is not in $\hat{\mathcal{V}}_{target}$. Because of these missing arguments, this will not be a valid new theorem. We note that although the extraction algorithm can be implemented in a way such that it "auto-completes" the arguments by adding additional necessary nodes into the set of extracted nodes, we choose not to do so in order to make sure the submodule is entirely identified by REFACTOR.

Once the extracted nodes pass these checks, we perform a so-called standardization. Specifically, we replace all node names of leaf nodes with a canonical set of variable names allowed in Metamath such as wph, wps. This can be achieved by first obtaining arguments of the extracted component and replacing them with the canonical variable names. As seen in Figure 4 (c), we replace all leaf node names wa with wps.

After standardization, we simply feed all the node names of the extracted component into the verifier we have described above to determine whether it is a valid theorem. For example, node names in (c) [wph, wps, wph, wn, wps, wn, hyp.1, hyp.2, 2th, con4bii] are fed into the verifier and we arrive at Figure 4 (d).

We note that verifying a proof after standardization is not always possible. Intuitively, what is true under a special context may not be true in the most general context. Consider an example in Figure 5 where the two parent nodes of blue node wi are not included in $\hat{\mathcal{V}}_{target}$ but in fact included in $\mathcal{V}_{target}$. Because of this, we need to replace wi with a canonical argument in Metamath such as wta. However, with this replacement, the arguments of syl5com will no longer be valid because it needs an expression with two wff variables in the node we substituted. Therefore, there will be no valid substitution and this proof tree prediction cannot be extracted as a new theorem. We discard the extracted components that cannot be verified after standardization and only consider the ones that can be verified as new theorems.

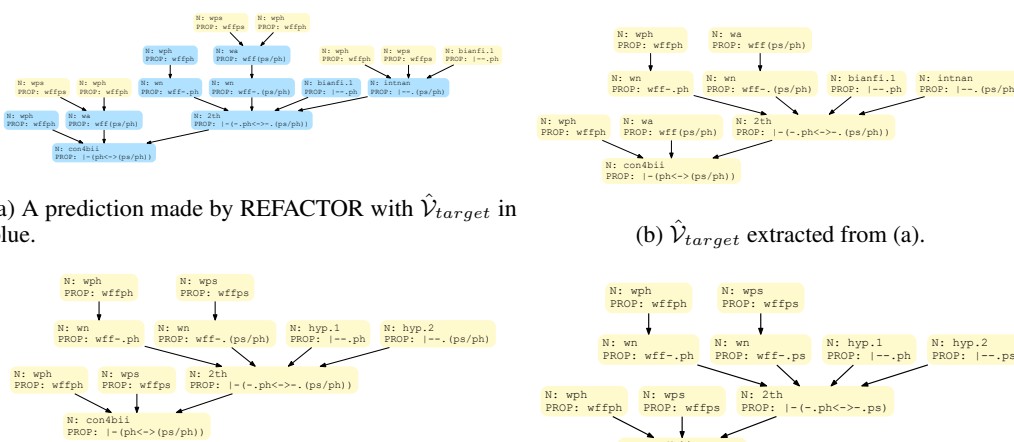

(a) A prediction made by REFACTOR with $\hat{\mathcal{V}}_{target}$ in blue.

(b) $\hat{\mathcal{V}}_{target}$ extracted from (a).

(c) $\hat{\mathcal{V}}_{target}$ extracted as in (b) with leaf node name and proposition replaced with canonical ones.

(d) A valid proof tree extracted and verified.

Figure 4: Visualization of theorem verification algorithm.

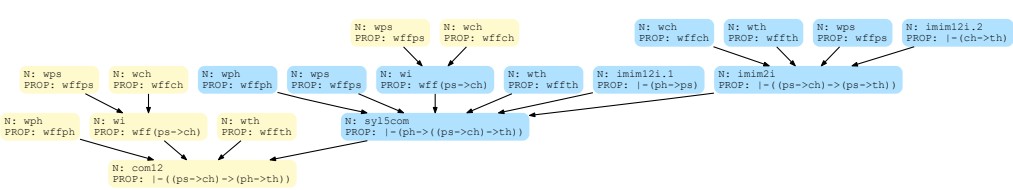

Figure 5: An example prediction that fails to be extracted as a new theorem due to no valid substitution plan in standardization. Specifically, the blue node wi cannot be substituted to a canonical argument allowed in Metamath while still keeping the proof tree valid.

## A.2 THEOREM EXPANSION

We discuss our theorem expansion algorithm in this section. An overview of the algorithm can be found in Algorithm 1. The algorithm takes input of two proof trees where the first proof tree applies a theorem in one of its steps and the second proof tree represents the proof of that theorem.

We explain our algorithm with the example from Figure 1. Specifically, proof tree $T$ corresponds to Figure 1 (b) and proof tree $T_s$ corresponds to Figure 1 (a). The theorem we want to expand is ali and we first obtain all its arguments using GetArguments function. We treat each theorem as a function and its arguments are the hypothesis of the theorem used to compute the conclusion. Consequently, the canonical arguments are wph, wps and ali.1. Next, we obtain contextual arguments, which are those specific hypotheses used in the context of the proof. Each hypothesis are represented by the entire subtree above each parent of $c$. Concretely, the contextual arguments of the ali node in (b) are wps, wch and [wph, wps, mp1i.a, mp1i.b, ax-mp]. Here, we use square bracket to enclose a subtree that has more than one node, which is treated holistically as the third contextual argument. Note that we can clearly see a one-to-one correspondence between the canonical arguments and the contextual arguments: (wph→wps, wps→wch and ali.1→[wph, wps, mp1i.a, mp1i.b, ax-mp]). We then simply replace all nodes in the proof tree of ali using this mapping. This gives us [wps, wch, wps, wi, wph, wps, mp1i.a, mp1i.b, ax-mp, wps, wch, ax-1, ax-mp]. We generate its proof tree representation with GetProof function. Finally we replace the subtree above ali with the new proof tree which in this case happens to be the entire proof of mp1i and this leads to the final expanded proof in Figure 1 (c).

---

**Algorithm 1** Theorem Expansion Algorithm Pseudocode

---

1: **procedure** EXPANSION
2:    **Input**: proof tree $P$ that uses theorem $t$ at node $c$.
3:    **Input**: proof tree of theorem $t$: $P_t$.
4:    canonicalArguments = GetArguments($P_t$)
5:    contextualArguments = [GetSubtree(p) for p in GetParents($c$)]
6:    allNodeNames = GetAllNodeNames($P_t$)
7:    $f$ : canonicalArguments $\rightarrow$ contextualArguments.
8:    $f(i^{th}$ element of canonicalArguments$) \triangleq i^{th}$ element of contextualArguments
9:    **for** each name $N \in$ allNodeNames **do**
10:      **if** $N \in$ canonicalArguments **then**
11:        replace $N$ with $f(N)$
12:    replacedProof = GetProof(allNodeNames)
13:    replace entire subtree above node $c$ with replacedProof
14:    **return** $T$

---

### A.3   THEOREM REFACTORING

In this section, we describe how we use newly extracted theorems to refactor the proof database of Metamath. Before proceeding, we first introduce how a basic refactor subroutine works. Consider the proof of `imim2i` in Figure 6 (a) and a new theorem extracted by REFACTOR in (b). The blue nodes in (a) can be refactored by the new theorem in (b) because they have two theorem application steps in common (`wi` and `a1i`). We can substitute arguments in (b) (`wffph`, `wffps`, `wffch`, and `|-(ph→ps)`) with arguments of blue nodes in (a) (`wffph`, `wffps`, `wffch` and `|-(ph→ps)`) respectively. After performing the substitution, we can replace all blue nodes in (a) with a single theorem application step of `new_theorem` along with its arguments. The refactored proof tree of `imim2i` is shown in Figure 6 (c).

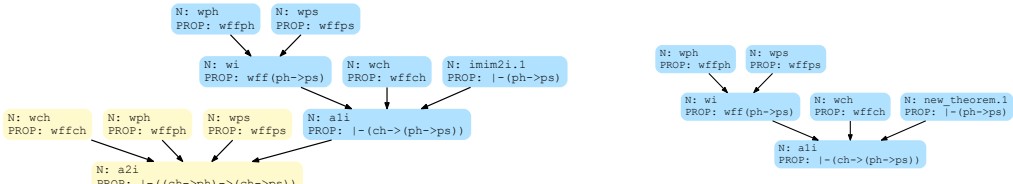

(a) Proof tree of theorem `imim2i` from `set.mm`. The blue nodes can be refactored by the new theorem in (b).

(b) A new theorem extracted by REFACTOR and can be used to refactor blue nodes in (a).

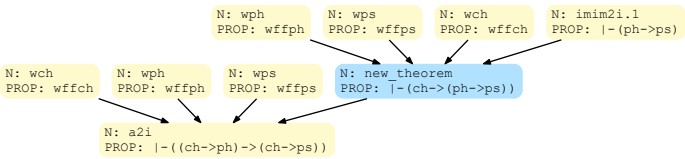

(c) Refactored proof tree of `imim2i` with new theorem highlighted in blue.

Figure 6: Visualization of a single refactoring operation. The theorem `imim2i` to be refactored is shown in (a), the new theorem used to refactor `imim2i` is shown in (b) and `imim2i` after refactoring is shown in (c).

We provide an overview of the refactoring algorithm in Algorithm 2. The algorithm aims to repeatedly perform the aforementioned refactoring subroutine on proof trees from existing Metamath database with each newly extracted theorem until no further subroutine can be performed. Our implementation refactors each proof tree in a post order traversal i.e. the leaves are attempted to be refactored before the root. This traversal is repeated whenever a refactor subroutine has been performed on the proof

tree by using a `while` loop. This is because once a theorem has been refactored, new theorems that are previously unable to refactor it might be applicable now. Different traversal order and which new theorem to refactor with first can potentially lead to different refactoring results and we leave this to future work.

---

**Algorithm 2** Refactoring Algorithm Pseudocode

---

1: **procedure** REFACTORING
2:    Proof trees $\{p^{(1)}, p^{(2)}, \cdots, p^{(m)}\}$ of theorems (to be refactored) in `set.mm`.
3:    Proof trees $\{q^{(1)}, q^{(2)}, \cdots, q^{(n)}\}$ of extracted new theorems.
4:    Generate post order node traversal $\{TR^{(1)}, TR^{(2)}, \cdots, TR^{(m)}\}$ for each proof tree in $\{p^{(1)}, p^{(2)}, \cdots, p^{(m)}\}$.
5:    **for** $i \in 1, 2, \cdots, m$ **do**
6:       **for** $j \in 1, 2, \cdots, n$ **do**
7:          **while** True **do**
8:             **for** all node $\in TR^i$ **do**
9:                match = RefactorSubroutine(node, $q^j$)
10:               **if** match == True **then**
11:                  Update post order node traversal $TR^i$
12:                  **goto** Line 7
13:             **break**
      **return** refactored proof trees $\{p_{ref}^{(1)}, p_{ref}^{(2)}, \cdots, p_{ref}^{(m)}\}$
14: **procedure** REFACTORSUBROUTINE
15:    **Input**: proof tree with root node $p$ that is matched against $q$.
16:    **Input**: proof tree $q$, an extracted new theorem.
17:    pSteps = GetSteps($p$)
18:    qSteps = GetSteps($q$)
19:    **if** pSteps != qSteps **then**
20:       **return** False
21:    **else**
22:       pArguments = GetArguments($p$)
23:       qArguments = GetArguments($q$)
24:       $f$ : qArguments $\rightarrow$ pArguments.
25:    $f(i^{th}$ element of qArguments$) \triangleq i^{th}$ element of pArguments
26:       refactoredTheorem = GetTheoremApplication($q$)
27:       replace node $p$ with refactoredTheorem
28:       **return** True

---

## B SYMBOLIC BASELINE PERFORMANCE

To demonstrate the significance of REFACTOR performance, we compare REFACTOR with a compression baseline similar to Vyskočil et al. (2010) that is purely symbolic and looks across multiple proofs. The baseline extracts the most frequently used theorems in the expanded dataset, similar to compression based approaches in related work. Specifically, for each node in a proof tree, the baseline extracts a valid theorem that uses all nodes above it. We take top $N$ most frequently extracted theorems where $N = 1923$ is the size of set of target theorems in our expanded dataset. We found that only 1.7% of them are actual theorems appeared in set.mm, compared to 19.6% achieved by REFACTOR.

## C THEOREM PROVING RESULTS WITH ORIGINAL HOLOPHRASM

With the original Holophrasm implementation, Holophrasm'16, REFACTOR also improved from the baseline as shown in Table 6. The absolute number of additional test theorems proved is 56 and slightly fewer than that from Holophrasm'20 results (75). This might be due to the fact that Holophrasm'16 is less well-trained and therefore limits the absolute improvement. The new theorem usage for the Holophrasm'16 case (43.6%) is higher than Holophrasm'20 (31.0%). We hypothesize this might be because more test theorems can be proved without using the new theorems for a better-trained prover.

Table 6: Number of test theorems proved comparison. The total number of test theorems is 2720. New theorem usage refers to the percentage of proved theorem that used new theorem at least once in its proof.

| Setting | Test Proof Found | New Theorem Usage |
|---|---|---|
| Holophrasm'16 (Whalen, 2016) | 412 | - |
| REFACTOR using Holophrasm'16 | **468** | 43.6% |
| Holophrasm'20 (Wang & Deng, 2020) | 557 | - |
| MetaGen (Wang & Deng, 2020) | 600 | - |
| REFACTOR using Holophrasm'20 | **632** | 31.0% |

## D EXTRACTED THEOREMS

We note that extracted theorems shown here are relatively simple but also used very frequently in both theorem refactoring and proving. Besides, we also extracted very sophisticated theorems that consist of hundreds of nodes (see Figure 2 (c) for the histogram distribution). We are actively looking into the implication of these theorems.

### D.1 FREQUENTLY USED THEOREMS IN REFACTORING

In Figure 7, we show the top 10 most frequently used new theorems in refactoring. Among them, two are extracted from the original set.mm and the rest are extracted from the expanded dataset. It is worth noting that although these theorems generally have fewer than 10 nodes each, they in total contribute to more than 78% of total number of nodes saved in refactoring, suggesting the pervasiveness and reusability of these extracted theorems in set.mm.

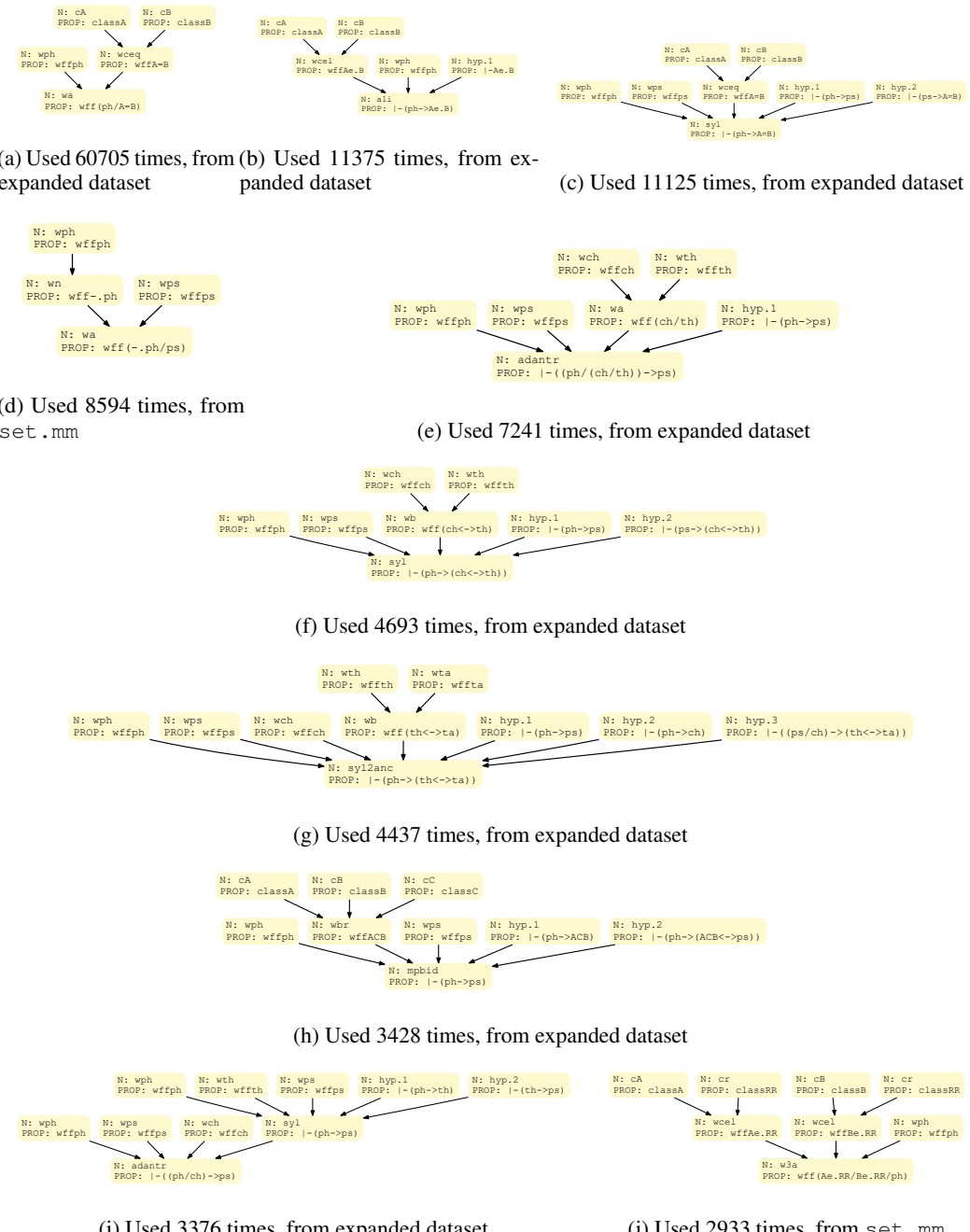

(a) Used 60705 times, from expanded dataset

(b) Used 11375 times, from expanded dataset

(c) Used 11125 times, from expanded dataset

(d) Used 8594 times, from `set.mm`

(e) Used 7241 times, from expanded dataset

(f) Used 4693 times, from expanded dataset

(g) Used 4437 times, from expanded dataset

(h) Used 3428 times, from expanded dataset

(i) Used 3376 times, from expanded dataset

(j) Used 2933 times, from `set.mm`

Figure 7: Top 10 most frequently used theorems in refactoring.

## D.2 FREQUENTLY USED THEOREMS IN THEOREM PROVING

In Figure 8, we show the top 10 most frequently used new theorems in theorem proving. All of them are extracted from the expanded dataset. It can be seen that the top 5 mostly used new theorems have fewer nodes than the other 5, suggesting these shorter new theorems are less proof specific and hence are used more frequently than those that are much longer and more applicable in niche proof context. Interestingly, there are two newly extracted theorems that show up in both Figure 7 and 8. The first one appears in both Figure 7 (b) and Figure 8 (c) and the second one appears in both Figure 7 (c) and Figure 8 (d). This overlap between frequently used theorems in refactoring and theorem proving further demonstrates the diverse utility of theorems we extracted.

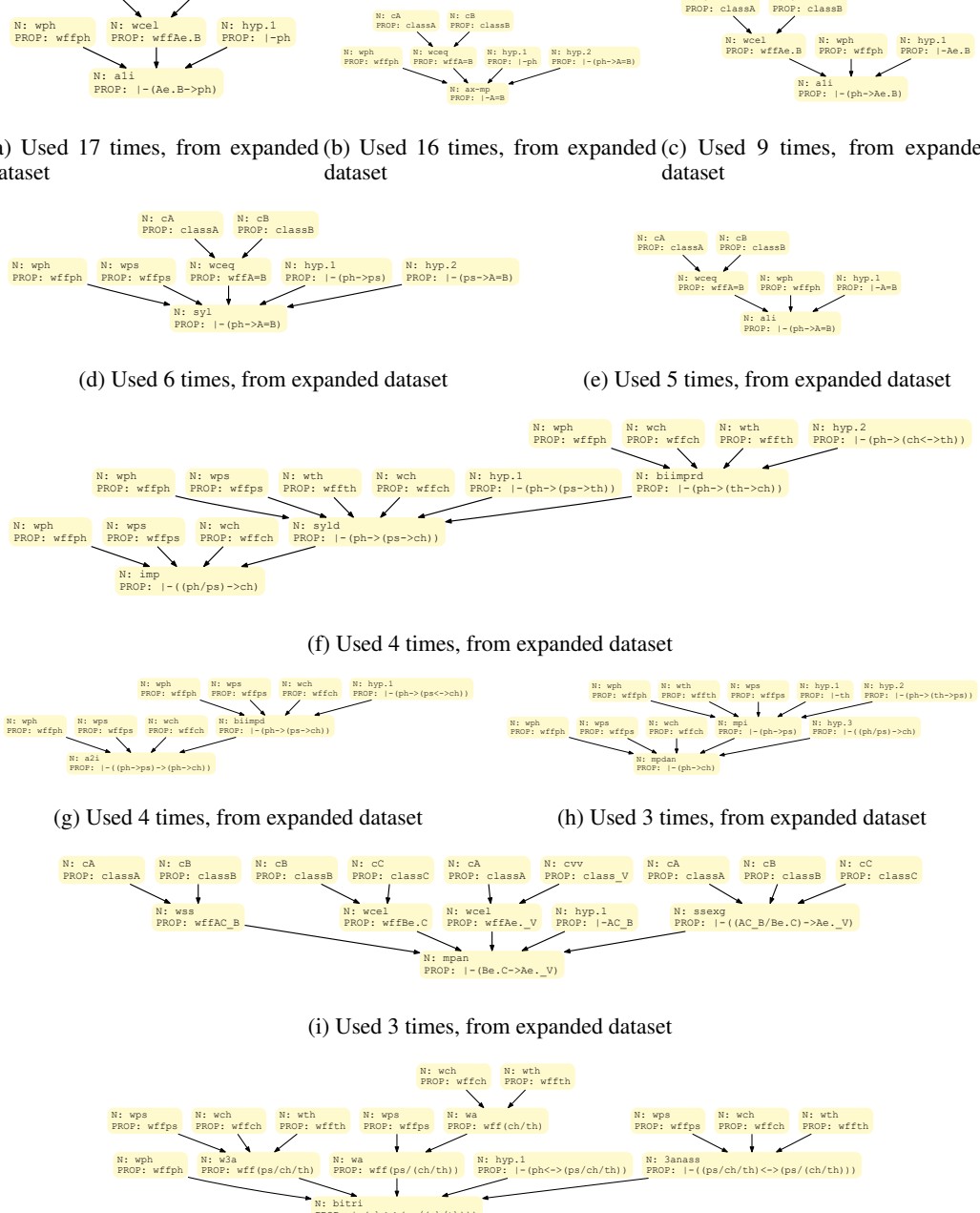

(a) Used 17 times, from expanded dataset

(b) Used 16 times, from expanded dataset

(c) Used 9 times, from expanded dataset

(d) Used 6 times, from expanded dataset

(e) Used 5 times, from expanded dataset

(f) Used 4 times, from expanded dataset

(g) Used 4 times, from expanded dataset

(h) Used 3 times, from expanded dataset

(i) Used 3 times, from expanded dataset

(j) Used 3 times, from expanded dataset

Figure 8: Top 10 most frequently used theorems in theorem proving.

