# OpenReview forum: "REFACTOR: Learning to Extract Theorems from Proofs"
_ICLR.cc/2024/Conference — ICLR 2024 poster_

### Official Review · Reviewer_QrE7 · 2023-10-29

**Soundness:** 3 good
**Presentation:** 3 good
**Contribution:** 2 fair
**Rating:** 8
**Confidence:** 4

**Summary:**

The paper proposes to extract a sub-tree from a complete proof tree to obtain new theorems. Given a human-written proof, it constructs training samples by expanding the theorems nodes with their proof tree. Then, this paper trains a graph neural network to embed the proof tree and classify whether a node is an expanded node to extract nodes for forming a theorem.

**Strengths:**

The paper improves the MetaGen performance by training it with the extra extracted theorems.

**Weaknesses:**

The necessity of training a neural network to extract sub-proof is not validated and explained. One can easily traverse the proof tree and obtain multiple sub-proof as theorems. This simple method can also surpass the proposed method in lots of aspects. For example, if it extracts all sub-proof, it might obtain all human-defined rules instead of 19.6% of them and more valid theorems than 1923. The experiments do not have a comparison with such baselines and cannot validate the effectiveness of the proposed method.

The paper does not give a detailed description in the main text of how to extract a theorem given the binary prediction results within a proof.

**Questions:**

- Why not fill some nodes with pre-defined rules to connect the positive nodes and obtain more valid theorems?

- How does the theorem extracted from the validation set occur/support the proofs from the test set?

- Do the positive and negative nodes balance in the training data?

- Can simple rule-based extraction methods, such as byte-pair encoding(BPE), improve the performance of ATP?

Post Rebuttal:
The response addresses my concerns and I have raised my rating.

---

> ### Author Response · Authors · 2023-11-15
> **Response to Reviewer QrE7**
>
> Thank you for your constructive feedback and we address individual points below.
>
> > The necessity of training a neural network to extract sub-proof is not validated and explained. One can easily traverse the proof tree and obtain multiple sub-proof as theorems. This simple method can also surpass the proposed method in lots of aspects. For example, if it extracts all sub-proof, it might obtain all human-defined rules instead of 19.6% of them and more valid theorems than 1923. The experiments do not have a comparison with such baselines and cannot validate the effectiveness of the proposed method.
>
> Thank you for your comment and we would like to provide some clarifications.
>
> If we traverse the proof trees and extract all possible sub-proof as theorems, the amount of theorems will be combinatorial to the number of nodes in the proof tree, which is not practical and useful for downstream tasks. Instead, we compared our method against a symbolic extraction method [1] where we keep the number of extracted theorems being the same (1923 in this case). This makes a fair comparison and the results are discussed in the last paragraph of page 7. The symbolic baseline only achieves a 1.7% accuracy compared to 19.6% by REFACTOR.
>
> Furthermore, our goal is not only extracting new theorems but also demonstrating their usefulness on downstream tasks. In Table 4, we show that with our newly extracted theorems (REFACTOR), the Holophrasm theorem prover can prove 75 more test theorems whereas the symbolic baseline improves Holophrasm only by 9 theorems.
>
> These results demonstrate that training a neural network to extract new theorems is more effective than symbolic baselines.
>
> > The paper does not give a detailed description in the main text of how to extract a theorem given the binary prediction results within a proof.
>
> Sorry for the confusion and missing it in the main text. We simply extract all the nodes whose output probability is greater than 0.5. We provided more details on theorem extraction in Appendix A.1. We will update our main text based on the Appendix in the final version of the paper.
>
> > Why not fill some nodes with pre-defined rules to connect the positive nodes and obtain more valid theorems?
>
> Thank you for the great suggestion! More valid theorems can indeed be obtained with the introduction of pre-defined rules. In this work, our goal is to show the possibility of just training a neural network to extract useful theorems. A hybrid approach that combines with rule-based methods is complementary to REFACTOR and we leave this as future work.
>
> > How does the theorem extracted from the validation set occur/support the proofs from the test set?
>
> Thank you for raising this point. To test the generalization ability of the neural network, we performed a target-wise split on the dataset. That is, the theorems to be extracted are non-overlapping for the training, validation and test set. We provide more detailed discussion on dataset creation in Section 5.1.
>
> > Do the positive and negative nodes balance in the training data?
>
> Thank you for the comment. Because of the presence of many large proof trees, the positive and negative nodes are not balanced. We did not explicitly balance the number of positive and negative nodes in the training data. The proof-level test accuracy of 19.6% attained by our best model suggests the model can still learn under unbalanced data.
>
> > Can simple rule-based extraction methods, such as byte-pair encoding(BPE), improve the performance of ATP?
>
> Thank you for the suggestion. In Table 4 and Section 5.6, we use a simple rule-based extraction method [1] optimized for compression objective to extract new theorems and investigate whether it can improve the theorem prover performance. Compared to the symbolic baseline, REFACTOR leads to proving 66 more test theorems, demonstrating the usefulness of the extracted theorems.
>
> [1] Vyskočil, Jiří, David Stanovský, and Josef Urban. "Automated proof compression by invention of new definitions." In Logic for Programming, Artificial Intelligence, and Reasoning: 16th International Conference, LPAR-16, Dakar, Senegal, April 25–May 1, 2010, Revised Selected Papers 16, pp. 447-462. Springer Berlin Heidelberg, 2010.

---

> > ### Author Response · Authors · 2023-11-19
> > **Follow up**
> >
> > Dear Reviewer QrE7,
> >
> > We thank you for your time and feedback, and would be happy to answer any further questions you may have before the discussion period ends. Please let us know if any issues remain and/or if there are any additional clarifications we can provide.
> >
> > If you are satisfied with our rebuttal, we would appreciate it if you could reconsider your score.
> >
> > Best regards,
> >
> > Authors

---

> > > ### Comment · Reviewer_QrE7 · 2023-11-21
> > >
> > > Thanks for the response. I am still concerned about the experiments.
> > >
> > > The paper first gives the accuracy and the number of extracted theorems. However, I don't think these two metrics solely can support the claim. Symbolic methods can easily surpass the proposed method by extracting all sub-proofs to obtain more theorems and 100% human-theorems recall. Limiting the theorem number to 1923 is unfair as the proposed method can only extract 1923 theorems while symbolic methods can extract much more. Thus, the accuracy and the extracted theorems number are not appropriate metrics.
> > >
> > > How much the extracted theorems can help prove is a more important metric. The paper shows the results in Table 4 but I still have some questions about the setting:
> > > - What is the set for training the REFACTOR, extracting the theorems, and testing the Holophrasm respectively?
> > > - What are the corresponding splits for the symbolic baseline?
> > >
> > > I think a proper setting is to train the REFACTOR on the training set, extract theorems from the valid to demonstrate its extraction ability and test the Holophrasm on the test set to show the usefulness of the extracted theromes. In this way, the REFACTOR will not memorize the groundtruth theorems, and the Holophrasm will not see the partial groundtruth proof.
> > >
> > > Looking forward to the response.
> > >
> > > I have another minor question: If the Holophrasm is trained with augmented theorems and the "correct" theorem is another human-proved theorem substituted into another proof, will the "incorrect" theorems be the actual new theorems that do not exist in the original dataset and eventually improve the Holophrasm.

---

> > > > ### Author Response · Authors · 2023-11-21
> > > > **Response to Reviewer QrE7**
> > > >
> > > > Dear Reviewer QrE7,
> > > >
> > > > Thank you for your valuable feedback and concerns. We are happy to provide more clarification.
> > > >
> > > > We agree with the reviewer that solely looking at accuracy and number of extracted theorems are not meaningful since we can extract all possible theorems in the corpus with symbolic baselines. What we did show with these two metrics is that, for the first time, a neural network could learn to extract human-written theorems that were synthetically expanded by us. The neural network could generalize to extract unseen theorems in the validation set (with 19.6% accuracy).
> > > >
> > > > Regarding the dataset splits, to ensure theorem proving performance are directly comparable to each other, we use the same test theorem split in the original Holophrasm paper. For both REFACTOR and symbolic baselines, we exclude these test theorems from the training and/or extraction phase.
> > > >
> > > > For REFACTOR, we train the model on a training set and report extraction accuracy (19.6%) on the validation set. We then let the model make predictions on both training and validation sets and extract the new theorems (~1923). The extracted theorems always differ from ground truth targets and by performing deduplication and standardization, we make sure the new theorems are not equivalent to existing theorems. For the symbolic baseline, it observes all proof trees in the training and validation set for optimizing the objective. The extraction accuracy is also evaluated on the validation set.
> > > >
> > > > By doing so, we made our best effort in reducing data contamination and having a fair comparison for theorem proving evaluation.
> > > >
> > > > Thank you for the minor question. We agree this could certainly happen in theory. In fact, even without augmented theorems, Holophrasm potentially could have seen proof steps from a training theorem that resembles or constitutes the proof of a test theorem. This could be due to the imperfection of human-written libraries and the low level nature of Metamath. Empirically, with the augmented theorems, we did not observe occurrences where the augmented theorems directly / trivially proved test theorems.
> > > >
> > > > Thank you for your insightful questions and scientific rigor. We look forward to answering any further questions you may have before the discussion period ends. Please let us know if any issues remain and/or if there are any additional clarifications we can provide.
> > > >
> > > > Best regards,
> > > >
> > > > Authors

---

> > > > > ### Comment · Reviewer_QrE7 · 2023-11-22
> > > > >
> > > > > Thanks for the clarification. It has addressed most of my concerns. One more question: How many theorems are extracted by the symbolic baseline for training Holophrasm?

---

> > > > > > ### Author Response · Authors · 2023-11-22
> > > > > > **Response to Reviewer QrE7**
> > > > > >
> > > > > > Dear Reviewer QrE7,
> > > > > >
> > > > > > Thank you for your question and we are glad to hear that our response addressed most of your concerns. To keep a fair comparison, we limited the symbolic baseline to also extract 1923 theorems and used them for training Holophrasm. Please let us know if there are any additional clarifications we can provide. If you are satisfied with our rebuttal, we would appreciate it if you could reconsider the score.
> > > > > >
> > > > > > Best regards,
> > > > > >
> > > > > > Authors

---

### Official Review · Reviewer_VAcB · 2023-10-30

**Soundness:** 3 good
**Presentation:** 2 fair
**Contribution:** 3 good
**Rating:** 5
**Confidence:** 4

**Summary:**

A scheme is proposed for learning to extract subtrees from proofs -- seen as computation trees -- that are expected to be useful elements to add to a library of theorems. The algorithm is trained on data created by unraveling function applications within human-written proofs and substituting the proof tree of the function being replaced, thus performing imitation learning on semi-synthetic data. The new theorems extracted in this way from a Metamath library are shown to yield shorter proofs (using proof-search algorithms applied to both the original and refactored collection of theorems).

**Strengths:**

- Good motivation and introductory examples. Main ideas are clearly explained.
- Very interesting method of learning to compress by training on artificially unraveled function applications.

**Weaknesses:**

- Framing and related work:
  - It is claimed a few times that proof refactoring "mimics what human mathematicians do" or the like, but this is not backed up. Is it your intuition or do you have evidence?
    - I am not convinced that compression of proofs is similar to human conjecture-making and definition-building behaviour. It certainly makes sense as a procedure for defining new abstractions, but it does not account for intrinsically motivated conjecture-making (guessing a theorem is true and then trying to prove it).
  - Section 3 misses essential mathematical background to theorem proving. The equivalence of proofs and programs / computation trees is stated and assumed without further comment, but it is not a trivial concept. The Curry-Howard correspondence and how it applies to Metamath deserves more discussion. See or cite, e.g., [P. Wadler, "Propositions as types"] for historical overview.
- Writing unclarities and bugs:
  - The example in Fig. 1 was hard to make sense of. (Note that the reviewer is familiar with Lean but not Metamath.) Can the last paragraph of section 3 explain it in plain(er) language, explaining what types of objects `ph`, `wffph`, etc. are and what the theorem means?
  - Incorrect use of "connected component" (several times on p.4). It is used to mean "connected subgraph", when in fact "connected component" has a different and very well-established meaning in graph theory.
  - Related, please also explain in a few words why connected subtree is not sufficient for validity. Note that issues related to bound variables would only be worse in more sophisticated proof systems -- this could be discussed as a limitation.
  - Misc.: Please check your citations (`\citep`/`\citet`).
- Evaluations are only on a relatively simple library and there isn't fair comparison to training-free compression schemes for extracting the new theorems.
- Also see questions below.

**Questions:**

- The GNN model seems to output a binary logit at each node. How do you extract the subtree from the predictions? I could not find discussion of the exact procedure in the text. The code does an argmax scheme; did you try sampling or other approaches?
  - Such a procedure seems to have an important limitation, which it can't produce a multimodal distribution over subtrees (which is a problem if the compressions on two different subtrees are mutually exclusive).
- Regarding the evaluation:
  - Could newly extracted theorems be equivalent to existing ones by symmetries? Does this affect the experimental claims?
  - Do you know for certain that the "human-written" proofs were written each proof from scratch? How does it affect the claims if humans copied chunks between the proofs?

---

> ### Author Response · Authors · 2023-11-15
> **Response to Reviewer VAcB (Part 1)**
>
> Thank you for your valuable and detailed review on our paper. We address individual points below.
>
> > It is claimed a few times that proof refactoring "mimics what human mathematicians do" or the like, but this is not backed up. Is it your intuition or do you have evidence? I am not convinced that compression of proofs is similar to human conjecture-making and definition-building behaviour. It certainly makes sense as a procedure for defining new abstractions, but it does not account for intrinsically motivated conjecture-making (guessing a theorem is true and then trying to prove it).
>
> Thank you for the question. The intuition comes from scenarios where we notice there is some part of the proof that is self-contained and does not relate closely to the rest of the proof. It seems clearer to separate it out into a lemma both better presentation, future use and compositional generalization.
>
> Our focus of the paper is not to fully capture what human mathematicians would do which involves iterative processes and refinement. We fully acknowledge human mathematicians perform a variety of actions such as conjecture making, definition building besides theorem abstraction from existing knowledge base. As a first attempt to capture partially what a human mathematician would do, we perform theorem expansion on human written corpus for imitation learning targets. We then learn to extract a modular component from the existing corpora. The extracted new theorems are shown to be useful in improving an off-the-shelf theorem prover performance as seen in Table 4.
>
> > Section 3 misses essential mathematical background to theorem proving. The equivalence of proofs and programs / computation trees is stated and assumed without further comment, but it is not a trivial concept. The Curry-Howard correspondence and how it applies to Metamath deserves more discussion. See or cite, e.g., [P. Wadler, "Propositions as types"] for historical overview.
>
> Sorry for missing the important background in establishing equivalence between these two concepts. We have updated Section 3 of our paper pdf in blue. We provide a very high level explanation to establish the equivalence between proofs and computation citing the related work. We would highly appreciate your feedback on our revision.
>
> > The example in Fig. 1 was hard to make sense of. (Note that the reviewer is familiar with Lean but not Metamath.) Can the last paragraph of section 3 explain it in plain(er) language, explaining what types of objects ph, wffph, etc. are and what the theorem means?
>
> We apologize for the confusion and below is a more detailed explanation for Figure 1 (a) theorem *a1i*. We will incorporate the explanation in the final version of the paper.
>
> The theorem proved in Figure 1 (a) states if *ph* is true, then *(ps->ph)* is true. Although this is rather low-level and possibly trivial, it is proven with the axiom of simplification along with the rule of Modus Ponens. To begin with, we start from *wffps* and *wffph*. These are expressions that state variable *ps* and *ph* are well-formed. Next, a theorem *wi* is invoked, which leads to *wff (ps->ph)*. This means *(ps->ph)* is also well-formed. With *wffps* and *wffph* shown on the top right corner of Figure 1 (a), an axiom called “the principle of simplification” can be invoked to arrive at *|-(ph->(ps->ph))*. The axiom essentially states the expression *a->(b->a)* is always true. In the last step, we use the rule of Modus Ponens. Specifically, it takes in 4 arguments: *wffph*, *wff(ps->ph)*, *|-ph* and *|-(ph->(ps->ph))*. The third argument *|-ph* comes from the hypothesis of the theorem (if ph is true) and the rest of arguments have been shown to hold from above. By applying the rule of Modus Ponens, we arrive at the goal *|-(ps->ph)*.
>
> > Incorrect use of "connected component" (several times on p.4). It is used to mean "connected subgraph", when in fact "connected component" has a different and very well-established meaning in graph theory.
>
> > Related, please also explain in a few words why connected subtree is not sufficient for validity. Note that issues related to bound variables would only be worse in more sophisticated proof systems -- this could be discussed as a limitation.
>
> Sorry for the incorrect terminology and we have updated the paper pdf. Since both hypothesis statement and theorem application are represented as nodes in the proof tree, it is possible to have an incorrect number of hypotheses (arguments) for a theorem (function) invocation. We agree with the reviewer that the issue could be worse in more sophisticated proof systems. One possible remedy is to automatically fill in the missing hypotheses during extraction.

---

> > ### Author Response · Authors · 2023-11-15
> > **Response to Reviewer VAcB (Part 2)**
> >
> > > Misc.: Please check your citations (\citep/\citet).
> >
> > Thank you for the note. We have updated our paper pdf with the correct citation formats in blue.
> >
> > > Evaluations are only on a relatively simple library and there isn't fair comparison to training-free compression schemes for extracting the new theorems.
> >
> > We would like to emphasize that the library is not simple. We experiment both REFACTOR and baselines on *set.mm*, the largest Metamath libraries consisting of 38k diverse theorems. As mentioned in Section 5.1, we also split the training, validation and test set by target theorem so that extraction targets are non-overlapping in the three sets.
> >
> > Furthermore, the symbolic compression baseline can observe the entire corpus to optimize for compression performance. The poor performance suggests that simply extracting the most frequently appeared subtree is not effective.
> >
> > Finally, our evaluation also focuses on downstream tasks such as theorem proving to investigate the usefulness of extracted theorems. REFACTOR leads to proving 66 more test theorems than the compression baseline, demonstrating the effectiveness of our approach.
> >
> > > The GNN model seems to output a binary logit at each node. How do you extract the subtree from the predictions? I could not find discussion of the exact procedure in the text. The code does an argmax scheme; did you try sampling or other approaches? Such a procedure seems to have an important limitation, which it can't produce a multimodal distribution over subtrees (which is a problem if the compressions on two different subtrees are mutually exclusive).
> >
> > Thanks for the great questions and sorry about the confusion. We simply extract all the nodes whose output probability is greater than 0.5. We provided more details on theorem extraction in Appendix A.1. Other approaches such as sampling and searching on the proof tree could potentially lead to better performance and we leave this as future work.
> >
> > We agree with the reviewer that producing a multinomial distribution over subtrees with autoregressive models such as transformers could further enhance the performance by extracting more theorems. We also discussed this in the last paragraph of Section 5.4 and Conclusion. In this work, our contribution focuses more on proposing the first proof-of-concept of theorem extraction using neural networks and showing the usefulness of extracted theorems.
> >
> > > Could newly extracted theorems be equivalent to existing ones by symmetries? Does this affect the experimental claims?
> >
> > The effect of equivalence to existing theorems has been minimized. When we extract new theorems, we perform a standardization process (discussed more in Appendix A.1) that removes duplicates among themselves and between the existing ones due to symmetry and variable name substitutions. Consequently, the numbers reported reflect the number of new theorems extracted. Furthermore, the newly extracted theorems are used frequently in the theorem prover evaluation and therefore we believe they are meaningful.
> >
> > > Do you know for certain that the "human-written" proofs were written each proof from scratch? How does it affect the claims if humans copied chunks between the proofs?
> >
> > Thanks for the point. We agree that due to the low-level nature of Metamath, it is unlikely humans write each proof from scratch. Although our focus is not on assisting human mathematicians writing proofs, we believe our idea of extracting modular and reusable theorems shown in REFACTOR could contribute to this goal by contributing to frequently used chunks.

---

> ### Comment · Reviewer_VAcB · 2023-11-18
> **Thank you for the responses**
>
> Thank you for the answers, clarifications, and corrections. In particular, I seem to have missed that Table 4 has results comparing the number of test theorems proved with your method compared to compression-based baselines like [Vyskočil et al.]. I updated the score. I see that this can be a useful method, but the simplistic and deterministic way of selecting a proof subtree (via the logit thresholding) seems like a major limitation, one that isn't present in compression-based schemes.

---

> > ### Author Response · Authors · 2023-11-18
> > **Thank You for Your Feedback**
> >
> > Dear Reviewer VAcB,
> >
> > Thank you very much for taking time to respond to our rebuttal. We are glad to hear that our response helped address your questions on comparing the usefulness of REFACTOR with symbolic baselines on theorem proving task. We fully acknowledge that the current theorem extraction algorithm is simple and can be enhanced with sampling mechanisms in future work. In this paper, our main contribution is to show the first proof of concept of training a neural network to extract useful theorems from mathematical proof trees.
> >
> > Best regards,
> >
> > Authors

---

### Official Review · Reviewer_WVCw · 2023-11-01

**Soundness:** 4 excellent
**Presentation:** 4 excellent
**Contribution:** 3 good
**Rating:** 8
**Confidence:** 4

**Summary:**

The paper proposes a pipeline to train neural networks to extract reusable theorems from mathematical proofs. It uses a graph neural network to take in a proof tree and make node-level classifications. It non-trivially extracts correct unseen theorems that humans would use 19.6% of the time. These theorems are then incorporated into the library to help in generating shorter proofs and potentially enhance the efficiency of baseline theorem provers.

More literature should be added in the related work section, for example, there is no related work mentioned after the year of 2021.
The author provides a symbolic baseline to demonstrate that 19.6% is non-trivial. The approach described in the symbolic baseline is intuitive and a fair comparison. The result provides confidence that this is an interesting line of work.

I believe the work is meaningful for automated proof simplification and generating better training dataset to improve baseline theorem prover performance. However, I’m concerned whether extracting “new theorems” from existing proofs is helpful to the automated theorem proving community, or the maths community in general.

Given that the metamath library is one of the largest databases, it could be possible that the 19.6% is the best possible result for the current REFACTOR pipeline and architecture. Even though 19.6% is a non-trivial result, it would be meaningful if we know whether the results can be improved by expanding the theorems even more and generating more training data.

**Strengths:**

- This is a novel problem and the first application of GNN to this problem to the best of my knowledge, the idea of theorem expansion is quite intuitive. The result is novel and could be helpful for proof simplification and generating better proof dataset.

- Potentially very impactful

**Weaknesses:**

The related work comparison can be improved.

**Questions:**

Do you plan to apply it to other math libraries, e.g., LEAN? If so, can your technique be lifted to that setting without much effort. Please provide justification, if your answer is YES.

---

> ### Author Response · Authors · 2023-11-15
> **Response to Reviewer WVCw**
>
> Thank you for your encouraging review and comments! We respond to your individual questions below.
>
> > However, I’m concerned whether extracting “new theorems” from existing proofs is helpful to the automated theorem proving community, or the maths community in general.
>
> Thank you for the question. We believe our work is still meaningful for the automated theorem proving community. In this work, we propose a novel way to expand proof trees and then create targets for training a neural network. The method along with the generated data from Metamath could be helpful to researchers who are interested in improving automated theorem prover performance. Besides, we extracted 1923 new theorems and as shown in Table 4, these newly extracted theorems are also directly used to prove test set theorems. Therefore, we believe this is a meaningful contribution.
>
> > Even though 19.6% is a non-trivial result, it would be meaningful if we know whether the results can be improved by expanding the theorems even more and generating more training data.
>
> Thank you for raising this point. We believe with a more sophisticated way of theorem expansion such as recursive expansion, we could create more diverse targets to further improve the performance.
>
> > The related work comparison can be improved.
>
> Thank you for the great suggestion and we have updated the Related Work section of our paper pdf in blue.
>
> > Do you plan to apply it to other math libraries, e.g., LEAN? If so, can your technique be lifted to that setting without much effort. Please provide justification, if your answer is YES.
>
> Yes. Our work can be lifted to other popular theorem proving environments such as Lean. As discussed in Section 1 and 3, our approach only makes the assumption of a simple substitution inference rule. Proof trees in other math libraries such as Lean and Isabelle can also be represented as trees and mathematically support the substitution of lemmas. Furthermore, thanks to the development of theorem proving playground in Lean (LeanDojo) [1], interacting with the proof trees has become more convenient. With the development of better language models, the features on each node can also be embedded with a better semantic meaning.
>
> [1] Yang, Kaiyu, Aidan M. Swope, Alex Gu, Rahul Chalamala, Peiyang Song, Shixing Yu, Saad Godil, Ryan Prenger, and Anima Anandkumar. "Leandojo: Theorem proving with retrieval-augmented language models." arXiv preprint arXiv:2306.15626 (2023).

---

### Official Review · Reviewer_QmHL · 2023-11-03

**Soundness:** 3 good
**Presentation:** 4 excellent
**Contribution:** 3 good
**Rating:** 8
**Confidence:** 4

**Summary:**

This paper proposes a deep learning method for extracting new proofs from human proofs. The task is formulated as a binary node classification of the nodes in the proof trees and a graph neural network is trained for this classification. Experiments on Metamath demonstrate the effectiveness of this approach. 1923 novel theorems are extracted from set.mm and these new theorems could greatly shorten human proofs. These new proofs could also be used to train the theorem prover and the performance of the Holophrasm prover is improved from 557/2720 to 632/2720.

**Strengths:**

1 The problem of extracting sub-proofs as standalone theorems is relevant and important. It can help discover new lemmas from existing proofs and compress the formal proof corpora.
2 The GNN-based approach is technically sound.
3 Experiments demonstrate the usefulness of the newly extracted theorems (1) they can be used to shorten human proofs (2) they can be used to improve the performance of theorem proving.

**Weaknesses:**

Metamath is a relatively simple formal mathematical language and less commonly used for advancing theorem proving compared to other formal provers like Isabelle and Lean. Although the proposed method may be applied to other provers, the implementation and the results of this paper doesn't contribute much to the advance of ATP directly.

**Questions:**

N/A

**Details Of Ethics Concerns:**

No ethics concern.

---

> ### Author Response · Authors · 2023-11-15
> **Response to Reviewer QmHL**
>
> Thank you for your thoughtful review and suggestions. We address individual points below.
>
> > Metamath is a relatively simple formal mathematical language and less commonly used for advancing theorem proving compared to other formal provers like Isabelle and Lean. Although the proposed method may be applied to other provers, the implementation and the results of this paper doesn't contribute much to the advance of ATP directly.
>
> Thank you for the comment. We fully acknowledge the fact that we implement REFACTOR on Metamath because of its simplicity and the inference rule (substitution). Our approach is also generally applicable to other formal systems such as Lean and Isabelle since proofs in these environments can also be represented as trees and mathematically support the substitution of lemmas.

---

### Author Response · Authors · 2023-11-15
**General Response to Reviewers**

We thank all the reviewers for their time and valuable comments. We appreciate that reviewers find the problem we are tackling **important** and **meaningful** (QmHL, WVCw), and our approach is **sound** and **interesting** with experimental results demonstrating its **usefulness** (QmHL, WVCw, VAcB).

As endorsed by several reviewers, the work makes important contributions to the field by demonstrating the possibility of training a neural network to extract useful theorems from mathematical proof trees. Specifically, we
- Formulate theorem extraction as a node-level binary classification problem and propose one straightforward way to create training datasets via theorem expansion.
- Demonstrate a GNN network trained on a theorem-split test set achieves 19.6% accuracy and extracts 1923 new theorems that are different from existing ones.
- Evaluate the usefulness of newly extracted theorems on improving downstream task performance such as theorem proving. Compared to symbolic baseline that optimizes for compression, REFACTOR leads to proving 66 more test theorems than the baseline.

We respond to each reviewer's valuable critiques in our individual responses. We hope to continue this valuable discussion during the discussion period!

---

### Meta-Review · Area_Chair_Zezi · 2023-12-09

**Metareview:**

This work considers the problem of extracting modular and reusable theorems from human proofs, with a deep learning approach. It receives mostly positive scores, where the reviewers think the problem is important and well motivated, and the method is interesting and clearly explained. Main concerns include 1. Evaluations on Metamath are relatively simple and not sufficient to demonstrate its effectiveness for theorem proving in general; 2. Some claims are not well grounded. AC agrees the problem of extracting new theorems from a large pool of proofs is interesting and this work is a nice proof-of-concept, and thus recommends acceptance as a poster. It could be consolidated by addressing above concerns.

**Justification For Why Not Higher Score:**

The probem is important and deserves further studies. The method is a nice proof-of-concept.

**Justification For Why Not Lower Score:**

The evaluations (on Metamath) are not sufficient to show its generalization to other more advanced and popular formal systems such as Isabelle and Lean.

---

### Decision · Program_Chairs · 2024-01-16

Accept (poster)